## RESEARCH ARTICLE

# Methylation-based signature to distinguish indolent and aggressive prostate cancer

Muheng Liao[1], Jace Webster[1], Amy Ly[1], Emily Rozycki[1] and Christopher A. Maher[1,2,3,*]

## ABSTRACT

Prostate cancer management faces significant challenges in distinguishing indolent from aggressive disease, particularly since most patients are intermediate-risk and therefore hinders the ability to recommend standardized treatment recommendations. Moreover, current prognostic tools including Gleason scoring and tumor staging demonstrate limited accuracy for predicting disease progression and tumor recurrence. DNA methylation serves as a stable epigenetic modification that directly regulates gene expression, making it an ideal biomarker for cancer prognosis. Therefore, this study leveraged whole-genome enzymatic methylation sequencing on 120 patients to develop a novel prognostic signature for aggressive prostate cancer progression. We analyzed 20,849 differentially methylated regions (DMRs) and employed multiple machine learning approaches to identify optimal biomarkers. This revealed a 14-region DNA methylation signature that can serve as independent prognostic prediction factors outperforming traditional clinical indices. Further, when combined into a risk score it achieved a clinically meaningful odds ratio. This methylation-based approach provides actionable information for treatment decisions and surveillance strategies, representing a significant advancement toward precision medicine in prostate cancer management through biologically informed risk stratification.

KEY WORDS: Prostate cancer, DNA methylation, Epigenome, Machine learning, Clinical prognosis

## INTRODUCTION

Prostate cancer (PCa) is one of the most common cancers in men in the United States, with an estimated 313,780 new diagnoses and 35,770 deaths in 2025 (Siegel et al., 2025). Although many patients exhibit prolonged indolence, around 25% progress (e.g. metastasis, regional recurrence) resulting in a 5-year survival rate between 29-35% (Siegel et al., 2025; Sherman et al., 2025). Due to the high discrepancy in patient outcomes, it is critical to identify accurate markers for identifying the transition from indolent to aggressive disease. Traditionally, identification of aggressive disease has been done by morphologically ranking tumors using Gleason scores (Gleason,

1966; Raychaudhuri et al., 2025). Trained pathologists will assign a score to the predominant and secondary patterns identified in a solid biopsy on a scale from 1-5 (with 1-2 being normal, healthy tissue and 3-5 being increasingly aggressive cancer tissue) and then report the score as the sum of the two regions. A sum of 6 is considered low grade or indolent disease, with higher sums representing increasingly aggressive disease states. While Gleason scores have been shown to correlate with patient outcomes (Hamdy et al., 2023; Swanson et al., 2021; Wu et al., 2025), certain drawbacks exist in prognosis. For example, studies have shown relatively poor concordance may exist due to the subjectivity of the system, especially in relatively indolent cases (Agosti and Munari, 2024). Furthermore, as indolent disease can be present for many years, it is not ideal to regularly collect solid tumor biopsies throughout the course of the disease.

To address existing shortcomings, biomarkers have been developed and applied to monitor PCa progression and are often used concordantly with Gleason scores. For example, detection of prostate-specific antigen (PSA) has been widely adopted for monitoring throughout all stages of PCa progression; although it has been observed to result in over-diagnosis and over-treatment, leading to unnecessary drops in the quality of life for patients (Loeb et al., 2014; Oh and Kang, 2025). While biomarker approaches methods based on genomic alterations are difficult due to the low frequency of recurrent mutations across patients at this stage of PCa (Al-Toubat et al., 2023), some gene-expression based methods have shown some promise for identifying aggressive disease (Farha and Salami, 2022). However, they could still suffer from requiring invasive collection of solid tumor biopsies (Wilson and Zishiri, 2024).

DNA methylation profiling has emerged as a promising approach for biomarker discovery in cancer (Yousefi et al., 2022; Shin et al., 2023). Methylation changes, epigenetic modifications involving the addition of methyl groups to cytosine bases in CpG dinucleotides that can alter gene expression without changing the underlying DNA sequence, can be measured objectively and do not suffer from the limited number of recurrent genomic mutations in PCa (Wang et al., 2022). A number of methylation changes in PCa diagnosis have already been well documented, such as hypermethylation of *GSTP1* and *APC* (Chen et al., 2013; Lin et al., 2001; Martignano et al., 2016). Prior work focused on building classification algorithms to differentiate between indolent and aggressive disease using Illumina methylation arrays on cohorts of patients with low- or high-grade disease only, but not specific intermediate ones (Toth et al., 2019; Wang et al., 2022). Importantly, the Illumina HumanMethylation450 BeadChip (450 K array) covers approximately 1.5% of all CpGs in the human genome (Fan et al., 2019), suggesting that there is a significant amount of information that is being systematically lost by using this approach. In contrast, enzymatic methylation sequencing (EM-seq) covers 26 million CpG sites; 57 times the coverage of the 450 K array (Vaisvila et al., 2021).

To address these challenges, first, we grouped PCa patients into low-, intermediate-, and high-risk based on Gleason core

[1]Department of Internal Medicine, Washington University School of Medicine, St Louis, MO 63110, USA. [2]Alvin J. Siteman Cancer Center, Washington University School of Medicine, St Louis, MO 63108, USA. [3]Department of Biomedical Engineering, Washington University McKelvey School of Engineering, St Louis, MO 63130, USA.

*Author for correspondence (christophermaher@wustl.edu)

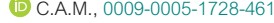 C.A.M., 0009-0005-1728-4614

and T stage. We performed a systematic and comprehensive whole genome methylation discovery between relatively indolent and aggressive PCa samples and subsequent verification on the remaining intermediate-risk cases that were 'kept out' of the discovery. Second, EM-seq was applied for methylation profiling across our unique cohort of 120 PCa patients with long-term clinical follow-up data. By comparing the low-risk group with the high-risk group, we identified 20,849 differentially methylated regions (DMRs), which captured previously understudied methylation characteristics of tumors with varying levels of malignancy in early-stage PCa. We demonstrated that these DMR markers could not only serve as independent prognostic prediction factors that outperformed traditional clinical indices but also combined into a risk score which achieved the highest odds ratio (OR). These findings provide new biomarkers for PCa patient classification, prognostic prediction, and long-term monitoring.

## RESULTS
### Whole-genome methylation analysis identifies differential methylated regions between low-risk and high-risk prostate cancer patients

To distinguish indolent from aggressive disease phenotypes, we conducted a retrospective cohort study (n=120) with long-term outcome data from patients archived in the Prostate Cancer

Biobank Network. We employed EM-seq for genome-wide methylation profiling since the samples were formalin-fixed, paraffin-embedded. EM-seq is optimized for degraded DNA, by avoiding the harsh bisulfite treatment used in conventional methods, thereby preserving fragment integrity while accurately detecting 5mC/5hmC modifications.

Our cohort (Table 1) is comprised of PCa patients between 46-77 years of age, with a median age of 62.5. Risk stratification was performed based on Gleason score (GS) and surgical T stage. Patients with GS=7 and surgical T stage ≤2c were classified as low-risk, whereas those with GS ≥9 or surgical T stage ≥3b were categorized as high-risk. The remaining patients were defined as intermediate-risk. Patients were categorized as progressing if they had documented tumor recurrence, PSA recurrence, lymph node or distant metastasis, and/or prostate cancer specific mortality. Notably, all low-risk patients were tumor progression negative, while 51 of 75 intermediate-risk patients and 21 of 25 high-risk patients were progression positive. These findings suggest a positive correlation between increasing risk stratification and progression rate.

Sample-level methylation profiling demonstrated that EM-seq covered approximately 26 million CpG sites (Fig. 1A,B). Each sample exhibited consistent proportions across genetic and CpG island annotation categories, confirming robust sequencing quality. Among detected CpG sites, 7.7% were located in promoters, 5.2%

**Table 1. Clinical characteristics of patients in this study**

| Characteristic | | Overall (n=120) | Low-risk (n=20) | Intermediate-risk (n=75) | High-risk (n=25) |
|---|---|---|---|---|---|
| Race | | | | | |
| | Black or African-American | 27 | 6 | 19 | 2 |
| | White or Caucasian | 93 | 14 | 56 | 23 |
| Age at surgery | | | | | |
| | Median (range) | 62.5 (46-77) | 56 (48-67) | 64 (46-76) | 61 (47-77) |
| Gleason score | | | | | |
| | 3+4 | 38 | 12 | 26 | 0 |
| | 4+3 | 34 | 8 | 24 | 2 |
| | 3+5 | 5 | 0 | 5 | 0 |
| | 4+4 | 2 | 0 | 2 | 0 |
| | 4+5 | 40 | 0 | 18 | 22 |
| | 5+5 | 1 | 0 | 0 | 1 |
| T stage at clinical diagnosis | | | | | |
| | 1c | 81 | 13 | 51 | 17 |
| | 2a | 27 | 7 | 16 | 4 |
| | 2b | 7 | 0 | 4 | 3 |
| | 2c | 5 | 0 | 4 | 1 |
| T stage at surgery | | | | | |
| | 2a | 2 | 1 | 1 | 0 |
| | 2b | 1 | 0 | 1 | 0 |
| | 2c | 57 | 19 | 36 | 2 |
| | 3a | 24 | 0 | 23 | 1 |
| | 3b | 36 | 0 | 14 | 21 |
| Survival | | | | | |
| | Alive | 99 | 20 | 67 | 12 |
| | Dead of disease | 11 | 0 | 0 | 11 |
| | Dead of other cause | 10 | 0 | 8 | 2 |
| Tumor recurrence | | | | | |
| | Yes | 60 | 0 | 44 | 16 |
| | No | 60 | 20 | 31 | 9 |
| PSA recurrence | | | | | |
| | Yes | 75 | 0 | 39 | 6 |
| | No | 45 | 20 | 36 | 19 |
| Lymph node metastasis | | | | | |
| | Yes | 101 | 0 | 10 | 9 |
| | No | 19 | 20 | 65 | 16 |
| Tumor progression | | | | | |
| | Yes | 48 | 0 | 51 | 21 |
| | No | 72 | 20 | 24 | 4 |

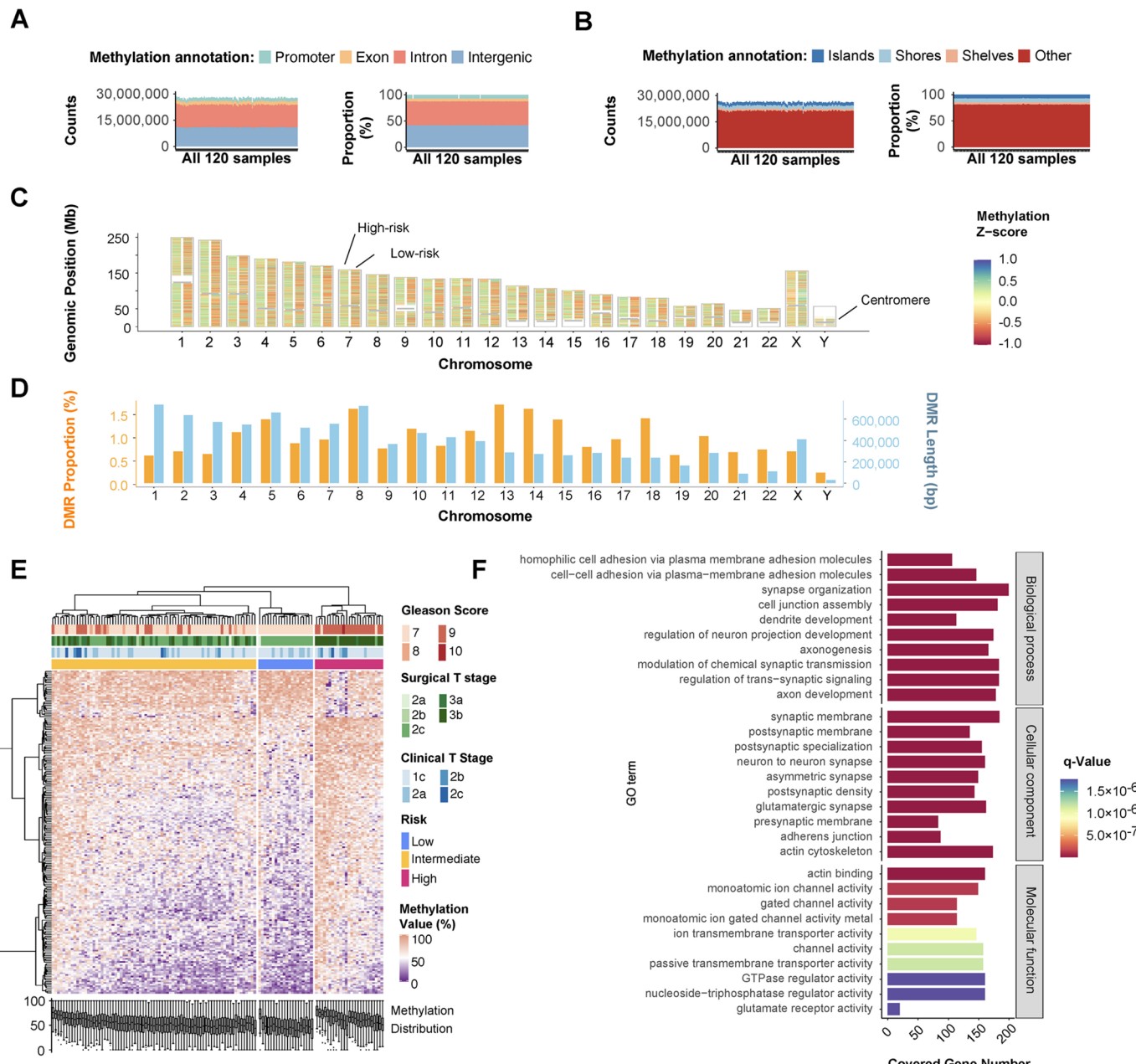

**Fig. 1. CpG sites kept consistent distribution in sample-level while derived DMRs demonstrated different methylation levels between each risk group.** (A) Genetic location covered by CpG cites in 120 samples, including promoters, exons, introns, and intergenic regions. (B) Methylation location covered by CpG cites in 120 samples, including CpG islands, shores, shelves, and others. (C) Physical location of DMRs on each chromosome with comparison of low-risk (right column) and high-risk (left column). The grey stick represented the centromere. (D) The length and proportion of DMRs on each chromosome. (E) Unsupervised hierarchical clustering of the top 200 most variable DMRs and 120 samples with clinical diagnosis under each risk group. Original methylation values $\left( \frac{reads\ of\ unmodified\ cytosine}{all\ reads} \times 100\% \right)$ instead of Z-scores were applied. (F) Gene ontology (GO) analysis of genes associated with all DMRs with a decrease false discovery rate (FDR) order in each category, biological process, cellular component, and molecular function.

in exons, 45.5% in introns, and 41.6% in intergenic regions (Fig. 1A). Similarly, within CpG island annotations, 7.7% were located on islands, 7.2% on shores, and 4.2% on shelves (Fig. 1B).

To ensure comprehensive detection without missing any potentially valuable DMRs, methylation analysis between low- and high-risk groups was conducted using both DSS and DMRcate packages (Peters et al., 2021; Pidsley et al., 2022; Zhao et al., 2020). Initially, a total of statistically significant 256,154 DMRs were identified, with 253,518 detected by DSS and 159,918 by DMRcate. 137,282 of them overlapped between the two methods (Fig. S1). Following the

criterion of absolute methylation difference higher than 10%, only 20,849 DMRs were kept as the valid DMR for further research. The overall DMR position on genetic region and CpG island matched with CpG sites (Fig. S2). Moreover, 91.53% of DMRs were only detected by EM-seq (Fig. S3) and were not covered within the Illumina 450 K array or Illumina Infinium MethylationEPIC Beadchips (EPIC array).

The chromosomal distribution of DMRs was summarized in Fig. 1C. The left columns represented the mean methylation Z-score of high-risk compared with low-risk in the right column for each chromosome. Analyzing DMR total length and coverage proportion

on each chromosome (Fig. 1D), Chr1 had the longest total DMR length whereas ChrY had the shortest total DMR length. Chr13 had the highest coverage proportion, while ChrY had the lowest.

The methylation profiling of the top 200 DMRs with the greatest absolute differences was presented in Fig. 1E, illustrating distinct methylation patterns between low- and high-risk groups. Two groups were well separated and clustered. While the intermediate-risk patients might have the feature of either group, it implied the possibility and necessity of introducing subgroups.

To explore biological significance, gene ontology (GO) enrichment analysis was performed on all DMRs (Fig. 1F), highlighting the top 10 most significant biological processes (BPs), cellular components (CCs), and molecular functions (MFs). Enriched BPs were primarily associated with cell adhesion and neuron-related terms, such as 'membrane adhesion', 'cell junction', 'axonogenesis', and 'synaptic transmission'. These findings aligned with CC terms like 'synaptic membrane', 'neuron synapse', and 'adherens junction'. For MFs, 'actin binding' and 'channel activity' were prominently enriched. These results reflected the differences in cell motility and organ morphogenesis between aggressive and indolent prostate tumors (Chen et al., 2023; Clark et al., 2023; Fu et al., 2024). Gene Set Enrichment Analysis (GSEA) further identified two significant gene sets: (a) genes transiently induced by EGF promoting cell-cycle progression; and (b) signal transmission across the membrane through G-protein activation enhancing the exchange of GDP for GTP on the alpha subunit of a heterotrimeric G-protein complex (Fig. S4).

### Development of a prognostic methylation signature

To establish a methylation signature of tumor progression for prognosis prediction, features were selected and filtered (Fig. 2A) from four machine learning models: Elastic Net Regression (ENR), XGBoost, Random Forest, and logistic regression with LASSO penalty. The initial set of 20,849 DMRs identified between low- and high-risk groups was applied to ENR for rapidly reducing dimensionality. Threshold optimization yielded approximately 100 DMRs per iteration across 50 runs (Fig. S5), generating 1213 candidate DMRs. Eighty-one DMRs were consistently identified by both XGBoost and Random Forest. These overlapping candidates underwent logistic regression with LASSO regularization to address multicollinearity, resulting in a final 14-DMR signature (Table 2). All of them located beyond the coverage 450 K or EPIC array.

Five DMRs were associated with protein-coding genes ABCC5, INPP4B, ALDOB, MAML2, and CLEC12A, respectively. Meanwhile, other three matched with long non-coding RNAs (lncRNAs), and the remaining six DMRs were not linked to any known genes.

Additional DMRs were identified in established PCa biomarker genes, including APC and GSTP1 (Figs S6,7) (Chen et al., 2013; Lin et al., 2001; Martignano et al., 2016). These regions displayed distinct methylation patterns between high-risk and low-risk groups, similar to chr9_101433398_101433510 (Fig. 2B), validating known methylation targets. Consistent methylation patterns were observed across all DMRs for each clinical outcome (Fig. 2C). Statistical assessment revealed that all 14 DMRs showed significantly different methylation levels between patients with and without tumor progression (Fig. S8).

### The 14-DMR signature and associated risk-score indicate the risk of tumor progression

A risk score was developed using logistic regression coefficients derived from low- and high-risk group classifications (Formula S1). Positive risk scores predominantly corresponded to cases with tumor progression, while negative scores indicated progression-free outcomes (Fig. 3A). Risk stratification demonstrated a clear dose-response relationship, with higher risk categories exhibiting elevated scores and increased progression rates.

### OR, CI is 95% confidence interval

The predictive performance of the risk score, 14-DMR signature, and established clinical parameters (T stage and GS) was evaluated using logistic regression (Fig. 3B, Table 3). In univariate analysis, all variables except GS achieved statistical significance ($P<0.05$). Tumor T stage demonstrated the strongest association with progression risk (OR=4.297, $P<0.001$, 95% CI: 1.992, 9.668). Eleven DMRs showed significant hypermethylation associations with progression, while three DMRs (chr9_1444688_1444802, chr1_27179908_27180212, and chr12_9968250_9968453) exhibited opposite effects. Then multivariate analysis identified seven independent predictors: the composite risk score and six individual DMRs (chr1_27179908_27180212, chr3_95125888_95125987, chr3_183997456_183997736, chr9_104670039_104670489, chr11_96194062_96194260, and chr18_52123037_52123163). Among these predictors, the risk-score demonstrated the strongest overall association with progression (OR=1.345, 95% CI: 1.033-1.957, $P=0.057$). Each unit increase corresponded to a 34.5% elevation in progression likelihood. Within the DMR signature, chr3_183997456_183997736 showed the most significant relationship with tumor advancement (OR=1.194, 95% CI: 1.058-1.419, $P=0.014$). Each percentage point rise in methylation at this locus enhanced progression probability by 19.4%. Conversely, chr1_27179908_27180212 exhibited a protective effect with the lowest odds ratio (OR=0.809, 95% CI: 0.675-0.909, $P=0.004$). Higher methylation levels at this region reduced progression risk by 19.1% per percentage point.

### Methylation signature demonstrated robust applicability in predicting prostate cancer progression

Principal component analysis of the 14-DMR signature across all 120 samples demonstrated effective separation of low- and high-risk groups, as well as stratification by tumor progression status and other clinical outcomes (Fig. 4A, Fig. S9). The analysis revealed potential molecular subtypes within the intermediate-risk group, suggesting heterogeneity not captured by traditional risk classification.

While the signature was discovered from low- and high-risk patients, we verified it on intermediate-risk patients and observed a high predictive accuracy for progression prediction (AUC=0.92). It substantially outperformed traditional clinical indices GS (AUC=0.65) and T stage (AUC=0.56, Figs 2C and 4B). Tumor recurrence was also predicted more effectively using the methylation signature, with an AUC of 0.84, compared to other indices (Fig. 4C). For tumor recurrence prediction, the methylation signature maintained superior performance (AUC=0.84) compared to conventional parameters (Fig. 4C). However, predictive accuracy for PSA recurrence, lymph node metastasis, and mortality was comparable to traditional indices (Fig. S10), likely reflecting the lower frequency of these events in the cohort.

### DISCUSSION

Prostate cancer patient management is hindered by the inability to reliably distinguish indolent and aggressive progression to avoid over- or under-treatment. Current prognosis approaches, including GS, checking T state, and PSA monitoring, are limited by subjectivity, invasiveness, and poor specificity (Loeb et al., 2014;

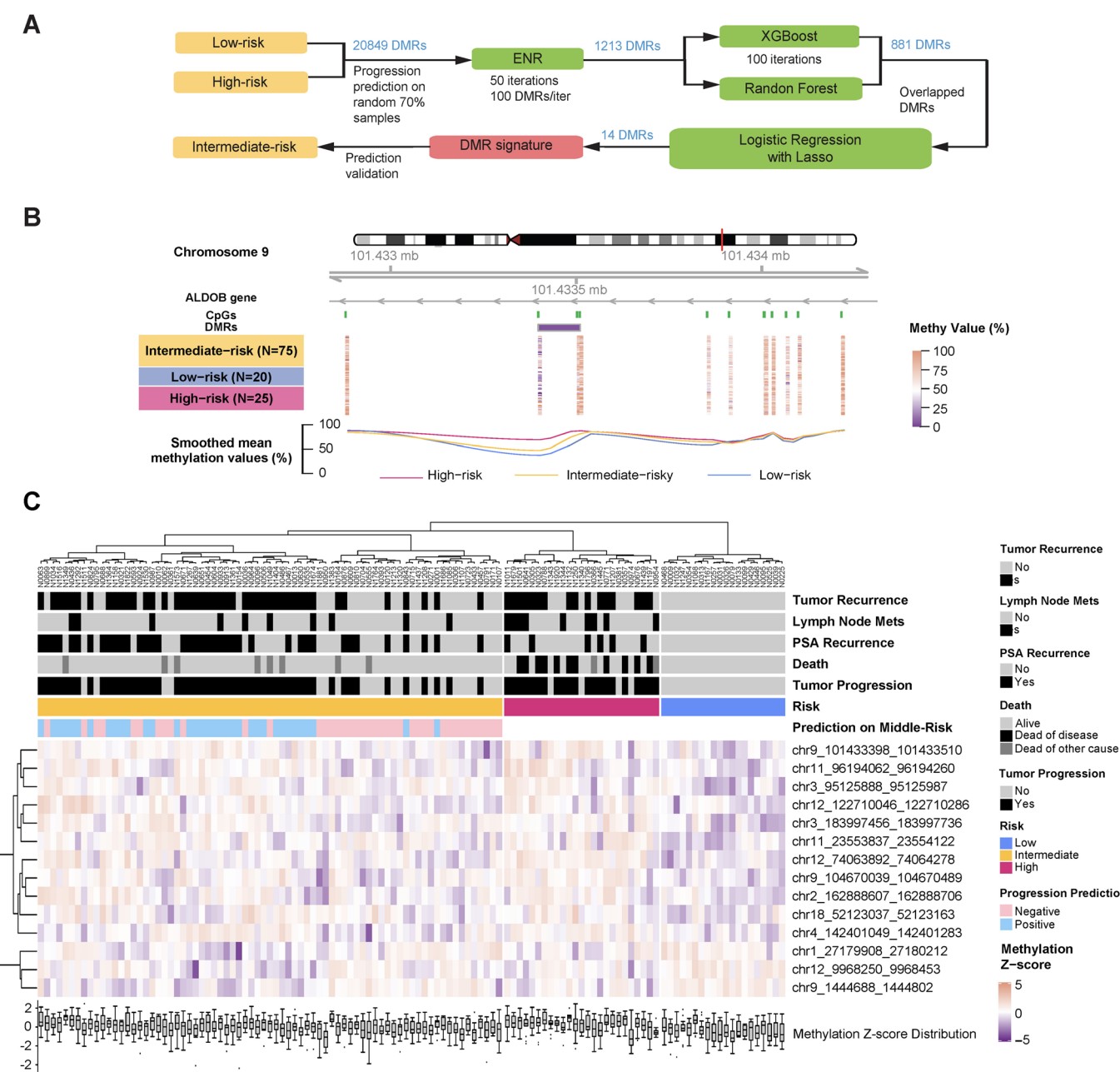

**Fig. 2. DMR signature of tumor progression recognized by machine learning algorithms from low-risk and high-risk samples could well serve for the prediction on intermediate-risk.** (A) The workflow of DMR signature (pink box) extraction with data grouping (yellow boxes) and algorithm (green boxes) details. (B) Genome browser showing the methylation level of typical DMR chr9_101433398_101433510 in each sample and risk group. Low-risk (light blue), intermediate-risk (orange), and high-risk (deep pink). (C) Unsupervised hierarchical clustering of final 14-DMR signature and 120 samples with clinical outcomes under each risk group. Samples marked in light green meant that tumor progression is unlikely to be experienced under the prediction of 14-DMR signature, while samples marked in light red were the opposite.

Oh and Kang, 2025). Although DNA methylation biomarkers offer advantages through their stability and direct relationship to gene regulation, typical studies using array-based platforms (Illumina 450 K or EPIC arrays) interrogate fewer than 2% of genomic CpG sites, potentially overlooking critical regulatory regions (Fan et al., 2019).

This study leveraged whole-genome EM-seq, which profiles 20-50-fold more CpG sites than conventional arrays and preserves DNA integrity through enzymatic conversion, making it suitable for low-input clinical samples (Vaisvila et al., 2021). Our systematic analysis identified 20,849 DMRs, with 91.53% detectable exclusively

through EM-seq, creating methylation signatures that outperformed traditional clinical indices. While previous studies focused on protein-coding regions including established markers GSTP1 and APC (also identified here), our genome-wide approach revealed novel DMRs predominantly in intergenic and intronic regions. These understudied regions demonstrated enrichment in pathways associated with tumor invasion and metastasis, including cell adhesion, synaptic signaling, and actin binding, highlighting their clinical relevance to aggressive disease progression (Kim et al., 2023).

Derived from all DMRs, we developed a methylation signature to predict aggressive PCa progression. The systematic approach

**Table 2. Methylation information and associated genes for the 14-DMR signature**

| DMR | Chromosome | Start | End | Width | Mean methylation value in high-risk | Mean methylation value in low-risk | Differential | Gene ID | Gene symbol | Gene biotype |
|---|---|---|---|---|---|---|---|---|---|---|
| chr1_27179908_27180212 | chr1 | 27179908 | 27180212 | 304 | 0.687 | 0.791 | −0.104 | | | |
| chr2_162888607_162888706 | chr2 | 162888607 | 162888706 | 99 | 0.701 | 0.579 | 0.122 | | | |
| chr3_95125888_95125987 | chr3 | 95125888 | 95125987 | 99 | 0.853 | 0.720 | 0.133 | ENSG00000239589 | LINC00879 | lncRNA |
| chr3_183997456_183997736 | chr3 | 183997456 | 183997736 | 280 | 0.743 | 0.618 | 0.125 | ENSG00000114770 | ABCC5 | Protein coding |
| chr4_142401049_142401283 | chr4 | 142401049 | 142401283 | 234 | 0.821 | 0.715 | 0.106 | ENSG00000109452 | INPP4B | Protein coding |
| chr9_1444688_1444802 | chr9 | 1444688 | 1444802 | 114 | 0.766 | 0.886 | −0.121 | | | |
| chr9_104670039_104670489 | chr9 | 104670039 | 104670489 | 450 | 0.818 | 0.675 | 0.143 | | | |
| chr9_101433398_101433510 | chr9 | 101433398 | 101433510 | 112 | 0.733 | 0.571 | 0.162 | ENSG00000136872 | ALDOB | Protein coding |
| chr11_96194062_96194260 | chr11 | 96194062 | 96194260 | 198 | 0.504 | 0.386 | 0.118 | ENSG00000184384 | MAML2 | Protein coding |
| chr11_23553837_23554122 | chr11 | 23553837 | 23554122 | 285 | 0.794 | 0.599 | 0.195 | | | |
| chr12_9968250_9968453 | chr12 | 9968250 | 9968453 | 203 | 0.763 | 0.873 | −0.110 | ENSG00000172322 | CLEC12A | Protein coding |
| chr12_74063892_74064278 | chr12 | 74063892 | 74064278 | 386 | 0.869 | 0.736 | 0.133 | ENSG00000251138 | LINC02882 | lncRNA |
| chr12_122710046_122710286 | chr12 | 122710046 | 122710286 | 240 | 0.695 | 0.578 | 0.117 | ENSG00000256249 | | lncRNA |
| chr18_52123037_52123163 | chr18 | 52123037 | 52123163 | 126 | 0.651 | 0.531 | 0.120 | | | |

integrated ENR for redundancy elimination, XGBoost and Random Forest for consensus feature selection, and Lasso regression for collinearity removal. This methodology yielded a robust 14-DMR signature that minimized statistical bias while maximizing predictive performance.

Biological exploration noticed five DMRs mapping to protein-coding genes (ABCC5, INPP4B, ALDOB, MAML2, and CLEC12A) with established oncogenic roles. ABCC5 facilitates prostate cancer advancement and enzalutamide resistance through the CDK1-

mediated AR Ser81 phosphorylation pathway (Ji et al., 2021). INPP4B inactivation in primary prostate cancers and preneoplastic lesions promotes the transition from androgen-regulated differentiation to proliferation (Hodgson et al., 2011). Reduced ALDOB expression correlates with poor clinical prognosis (Ning et al., 2021; Zhao and Xu, 2023). MAML2, through CRTC1-MAML2 fusion, serves as a major oncogenic driver in mucoepidermoid carcinoma (Chen et al., 2021; Musicant et al., 2021). While CLEC12A is primarily associated with leukemia, it

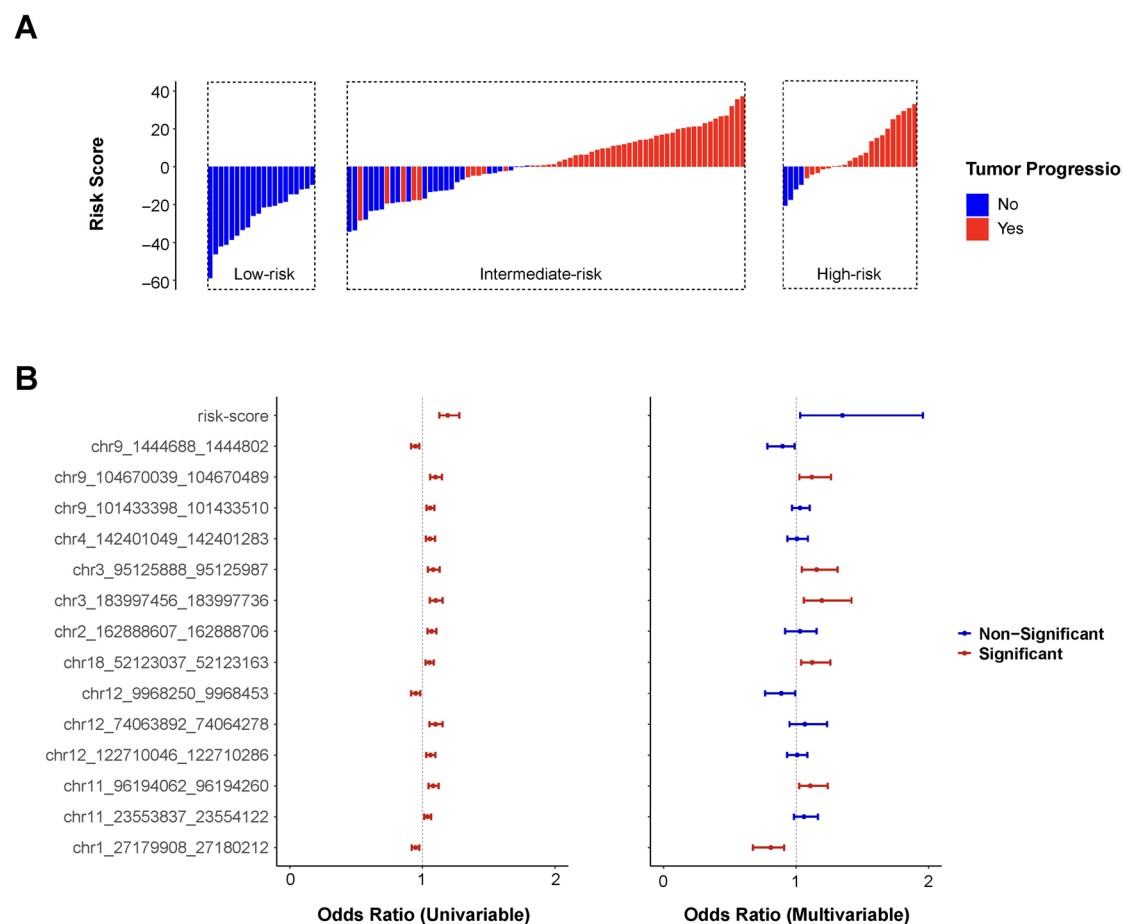

**Fig. 3. Risk-score derived from 14-DMR signature worked as an independent indicator for prognosis prediction of tumor progression.** (A) The distribution of risk-score in each risk group with color mark of tumor progression (red) or not (blue). (B) Odds ratios and relevant statistical significance (*P*<0.05, red; *P*>0.05, blue) of risk-score each DMR in the 14-DMR signature from univariate and multivariate analysis.

**Table 3. Odds ratios of each DMR in univariate and multivariate analysis**

| DMR | Gene | Univariate | | | Multivariate | | |
|---|---|---|---|---|---|---|---|
| | | P-value | OR | CI | P-value | OR | CI |
| chr1_27179908_27180212 | - | **<0.001** | 0.949 | 0.919-0.976 | **0.004** | 0.809 | 0.675-0.909 |
| chr2_162888607_162888706 | - | **<0.001** | 1.069 | 1.038-1.105 | 0.608 | 1.030 | 0.916-1.156 |
| chr3_95125888_95125987 | LINC00879 | **<0.001** | 1.081 | 1.042-1.129 | **0.011** | 1.155 | 1.044-1.314 |
| chr3_183997456_183997736 | ABCC5 | **<0.001** | 1.100 | 1.057-1.152 | **0.014** | 1.194 | 1.058-1.419 |
| chr4_142401049_142401283 | INPP4B | **<0.001** | 1.057 | 1.025-1.094 | 0.862 | 1.007 | 0.934-1.089 |
| chr9_1444688_1444802 | - | **0.001** | 0.947 | 0.914-0.977 | 0.057 | 0.897 | 0.783-0.988 |
| chr9_104670039_104670489 | - | **<0.001** | 1.099 | 1.058-1.148 | **0.029** | 1.119 | 1.025-1.264 |
| chr9_101433398_101433510 | ALDOB | **<0.001** | 1.058 | 1.031-1.090 | 0.346 | 1.030 | 0.970-1.102 |
| chr11_96194062_96194260 | MAML2 | **<0.001** | 1.081 | 1.047-1.123 | **0.028** | 1.108 | 1.023-1.238 |
| chr11_23553837_23554122 | - | **0.003** | 1.039 | 1.014-1.066 | 0.162 | 1.060 | 0.984-1.165 |
| chr12_9968250_9968453 | CLEC12A | **0.005** | 0.949 | 0.913-0.983 | 0.061 | 0.888 | 0.766-0.993 |
| chr12_74063892_74064278 | LINC02882 | **<0.001** | 1.098 | 1.053-1.152 | 0.315 | 1.066 | 0.950-1.234 |
| chr12_122710046_122710286 | - | **<0.001** | 1.061 | 1.029-1.099 | 0.831 | 1.008 | 0.933-1.086 |
| chr18_52123037_52123163 | - | **<0.001** | 1.052 | 1.024-1.084 | **0.015** | 1.121 | 1.038-1.260 |
| Risk-Score | - | **<0.001** | 1.190 | 1.126-1.279 | 0.057 | 1.345 | 1.033-1.957 |
| Gleason score | - | 0.403 | 1.379 | 0.653-2.963 | - | - | - |
| Tumor T stage | - | **<0.001** | 4.297 | 1.992-9.668 | - | - | - |

Statistically significant P-values are in bold.

has also been shown to sensitize breast cancer cells to artemisinin treatment by suppressing autophagy and inflammation (Chatterjee and Chatterji, 2022; Chatterjee et al., 2023).

During clinical prognosis, several traditional markers of prostate cancer have been widely applied. GS serves as the cornerstone for histological grading, while tumor T stage provides essential anatomical staging information (Cheng et al., 2012; Egevad et al., 2002). These parameters have been integral to clinical decision-making in NCCN Guidelines for decades. However, their limitations in accurately predicting disease progression, particularly in intermediate-risk patients, have been increasingly recognized. Several groups have proposed subdividing intermediate-risk prostate cancer into favorable and unfavorable categories based on factors such as primary GS, number of intermediate-risk features, and percentage of positive biopsy cores (Courtney et al., 2022; Zumsteg et al., 2013). These refined classifications improved prognostic discrimination for biochemical recurrence, metastasis, and mortality compared with the traditional three-tier model. Additionally, proper treatment also requires further stratification in the intermediate-risk group. For instance, prior works suggests that it is unlikely

that treatment intensification would meaningfully improve oncologic outcomes in favorable intermediate risk group (Berlin et al., 2019). However, over- or under-treatment could be a potential problem.

Our findings were consistent with and extend this concept at the molecular level. Within our cohort, the majority of patients (62.5%) fell into the intermediate-risk group, which historically exhibits the greatest uncertainty in treatment selection. The strong discriminative ability of our 14-DMR methylation signature effectively separated progressive from indolent cases within this heterogeneous population, paralleling the clinical distinction between unfavorable and favorable intermediate-risk disease.

In this situation, our 14-DMR signature demonstrated markedly superior predictive accuracy (AUC=0.92) and sharply contrasted with GS (AUC=0.65) and tumor T stage (AUC=0.55). This substantial improvement represents a clinically meaningful advancement. The intermediate-risk category made up 62.5% of this cohort and faced the greatest uncertainty regarding treatment selection and prognosis. The enhanced performance of our methylation signature highlights it potential clinical utility.

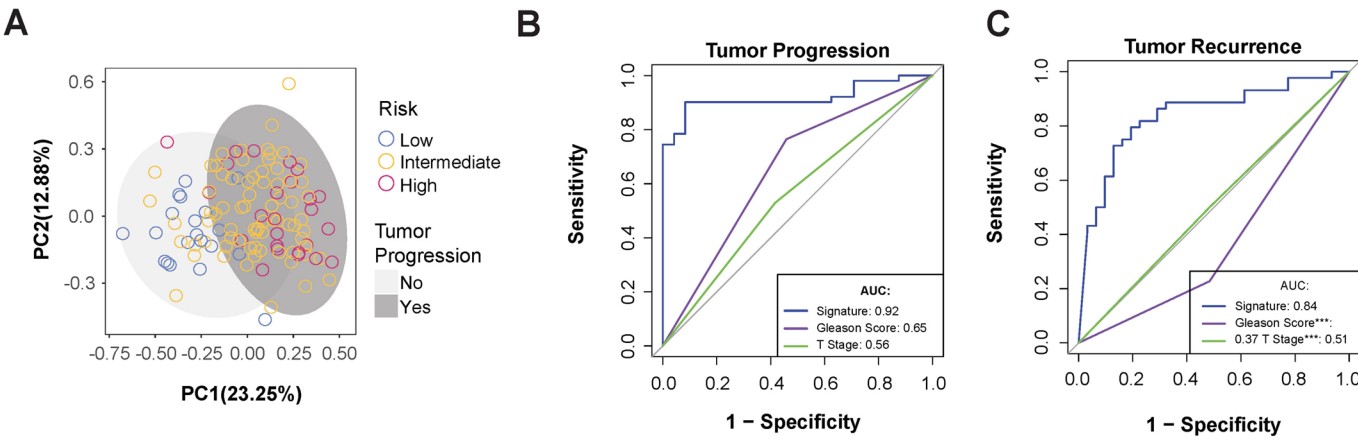

**Fig. 4. The DMR signature demonstrated robust applicability in predicting tumor progression.** (A) Principal component analysis of 14-DMR signature patients under 120 samples with tumor progression (deep grey) or not (light grey). Circle color referred to the risk group. Tumor progression (B) and tumor recurrence (C) ROC curve of prognostic prediction using 14-DMR signature (blue), GS (purple), and tumor T stage (green). The AUC value of each curve and the statistical significance (*0.05, **0.01, ***0.001) of differences compared to the signature were summarized in the lower right corner of the plots.

In this situation, our 14-DMR signature demonstrated markedly superior predictive accuracy (AUC=0.92) and sharply contrasted with GS (AUC=0.65) and tumor T stage (AUC=0.55). This substantial improvement represents a clinically meaningful advancement. The intermediate-risk category made up 62.5% of this cohort and faced the greatest uncertainty regarding treatment selection and prognosis. The enhanced performance of our methylation signature highlights it potential clinical utility.

The clinical utility of the methylation signature is particularly evident for tumor recurrence prediction (AUC=0.84), an outcome that traditional indices assess with limited precision. Tumor recurrence is a critical clinical endpoint that directly influences long-term patient outcomes and necessitates aggressive therapeutic intervention. Accurate risk identification could enable personalized treatment intensification and surveillance strategies. While GS and T stage provide valuable information for overall survival and biochemical recurrence, their predictive power for local recurrence has been insufficient for optimal clinical decision-making. However, the methylation signature had limited predictive power for PSA recurrence, lymph node metastasis, and death outcomes. This likely reflected imbalanced sample distributions during signature discovery. The relatively low incidence of these events in our cohort may have precluded adequate statistical power for detection of meaningful associations. This observation underscores the importance of larger, multi-institutional studies with extended follow-up periods.

To quantify tumor progression risk systematically, a composite risk score was developed via logistic regression. The score incorporated coefficients that weight individual DMR contributions based on their prognostic significance. Higher risk scores correlated strongly with increased likelihood of disease progression. This confirmed the validity of the score as an independent prognostic indicator and suggest that epigenetic information provides additive value beyond conventional risk stratification approaches.

Subsequent univariate and multivariate analyses provided robust statistical evidence. In univariate analysis, each DMR within the signature, risk-score, and T stage demonstrated clinical utility. In multivariate models, only six DMRs retained independent prognostic significance ($P<0.05$). Chr3_95125888_95125987, chr3_183997456_183997736, chr9_104670039_104670489, chr11_96194062_96194260, and chr18_52123037_52123163 and corresponding genes LINC00879, ABCC5, and MAML2 carried OR>1 indicating the association of hypermethylation and PCa progression. With OR<1, chr1_27179908_27180212 conversely implied hypomethylation in this region could be risky. These results were clinically meaningful and compared favorably to established prognostic biomarkers in prostate cancer.

Nevertheless, this study has several limitations. First, our risk stratification strategy was imperfect. The lack of follow-up time and PSA diagnosis might have led to the misclassification of a small number of patients into the current intermediate-risk category. Additionally, the methylation signature and risk-score held potential for non-invasive detection using cell-free DNA. However, this potential could not be validated within the present study due to the lack of corresponding fluid samples. Therefore, validating their feasibility in liquid biopsies would be the focus of a subsequent independent investigation.

Collectively, our results highlight the clinical utility of a DNA methylation signature for prognostic risk stratification in prostate cancer. Unlike purely clinicopathologic stratifications, our methylation signature captures underlying epigenetic alterations that may drive the biological divergence between these subgroups. Integrating such methylation-based markers with established frameworks could enable more precise risk stratification and facilitate personalized treatment planning for patients whose disease lies between low- and high-risk boundaries.

## MATERIALS AND METHODS

### Study population and sample collection

A total of 120 patients diagnosed with PCa were recruited from Prostate Cancer Biobank Network (PCBN). Sample collection details were as previously described (Zhao et al., 2020). Tissue specimens were collected through radical prostatectomy (RP) or transurethral resection of the prostate (TURP), without any prior treatment. All patients provided signed informed consent for the collection of specimens and subsequent genetic analyses.

Tumor T stage was determined both pre- and post-surgery, and GS were assigned based on histopathological analysis. Clinical outcomes were followed until loss connection, with tumor progression defined by the occurrence of any of the following: death, tumor recurrence, PSA recurrence, or lymph node metastasis. To stratify the degree of malignancy of the tumor, risk stratification criteria from NCCN Guidelines were utilized and subsequently optimized to achieve a greater contrast between low- and high-risk patients while maintaining balanced group sizes (Schaeffer et al., 2024). Patients with GS=7 and surgical T stage≤2c were classified as low-risk, those with GS≥9 and surgical T stage≥3b as high-risk, and the remaining patients as intermediate-risk. All samples were formalin-fixed and paraffin-embedded (FFPE) in 4°C environment.

### Enzymatic methylation sequencing experiment

Genomic DNA was extracted using Zymo Research *Quick*-DNA/RNA MiniPrep kit (Zymo Research, Irvine, CA, USA) and quantified by Qubit 2.0 fluorometer (Thermo Fisher Scientific, Waltham, MA, USA). Sample purity was determined by using the Nanodrop ND-1000 Spectrophotometer (Thermo Fisher Scientific, Waltham, MA, USA). And all samples were required to meet the following quality thresholds prior to library construction: DNA concentration ≥20 ng/μl, A260/A280 ratio between 1.8-2.0, and A260/A230 ratio ≥1.8.

EM-Seq libraries were prepared with 10-200 ng of DNA using the NEBNext Enzymatic Methyl-seq Kit and NEBNext Multiplex Oligos for Enzymatic Methyl-seq Unique Dual Index Primer Pairs (New England BioLabs, Ipswich, MA, USA) according to the manufacturer's protocol. Sample integrity and absence of degradation were verified via spike-in controls of Lambda DNA (Thermo Fisher Scientific, Waltham, MA, USA) and pUC19 plasmid (New England Biolabs, Ipswich, MA, USA). Library quantification was performed using the Qubit 2.0 fluorometer. Library quality and size distribution were assessed using an Agilent 2100 Bioanalyzer (Agilent Technologies, Santa Clara, CA, USA), followed by paired-end sequencing (2×150 bp) on an Illumina NovaSeq 6000 platform (Illumina, San Diego, CA, USA).

Raw reads were trimmed using TrimmGalore (v0.6.10). To remove methylation bias and remove low-quality sequences, an additional 10 bp of the 5′ ends of each read was removed. Duplicated reads were removed to reduce PCR bias. Then, they were aligned to the human genome reference (hg38) using Bismark (v0.24.0), with methylation calls derived by comparing cytosines at converted (T) versus protected (C) positions. Raw data had been deposited in Gene Expression Omnibus (GSE308050).

### DNA methylation analysis

DMRs were identified by analyzing methylation patterns between high-risk and low-risk groups using R packages DSS (v2.48) and DMRcate (v2.14.1) independently (Peters et al., 2015; Wu et al., 2013). The methylation level of each DMR was calculated as the cytosine frequency. Relevant data could be found in GSE308050. Valid DMRs were defined by an absolute methylation difference of >10% and an adjusted $P$-value of <0.05. The overlapping DMRs identified by both packages were defined by the results from DMRcate, as it provided more precise range detection under optimal settings. The distribution of these DMRs across chromosomes was summarized, and a heatmap was generated using the ComplexHeatmap package (v2.15.4) to visualize the methylation pattern of the top 200 DMRs with the greatest differences between low- and high-risk groups (Gu et al., 2016).

Genome and CpG island references were downloaded from UCSC Genome (https://hgdownload.soe.ucsc.edu/goldenPath/hg38/database/) (Perez et al., 2025). And relevant genetic and CpG annotation of all DMRs was conducted. The coordinates of CpG islands followed the classic definition from Gardiner-Garden & Frommer's work (Gardiner-Garden and Frommer, 1987). CpG shores were defined as 2 kb up- and downstream of islands, excluding the islands. Similarly, CpG shelves were defined as 2 kb up- and downstream of shores, excluding shores and islands.

DMRs were linked to genes based on direct overlap with the annotated gene region in the reference (Ensembl release 111) (Harrison et al., 2024). DMRs overlapping multiple genes were labeled with all overlapping genes, while not all DMR would be linked to a gene since they might also lay on the non-coding region.

## Methylation signature extraction

To identify key DMRs associated with prognosis outcomes, patients were categorized based on the presence or absence of tumor progression. And A machine learning pipeline was constructed to filter out the DMR signature, utilizing four algorithms: Elastic Net Regression, XGBoost, Random Forest, and Logistic Regression with Lasso penalty. The process began with Elastic Net Regression (glmnet, v4.1-8) as an initial filter (Friedman et al., 2010; Tay et al., 2023). 70% of the DMRs were randomly selected, then significant ones were collected in each of 50 iterations by adjusting the λ value ($10^{-30}$-10) to mitigate statistical bias. Next, the remaining DMRs were evaluated by XGBoost (xgboost, v1.7.5.1) and Random Forest (randomForest, v4.7-1.1) models (Breiman, 2001; Chen and Guestrin, 2016). Both models were applied to 100 iterations of each set of different parameters, selecting important DMRs from random 70% of the input. Each model was trained with a random 70% of the samples and tested on the remaining 30%. DMRs identified by both models with an accuracy >0.7 on the test set were considered as final candidates for the DMR signature. Logistic Regression with Lasso penalty (glmnet, v4.1-8) was then deployed to eliminate collinearity and select the final set of DMRs, resulting in a 14-DMR signature.

## Prognostic analysis

To validate the stratification ability of this signature on prognostic prediction, the intermediate-risk cohort were set as the validation cohort and predicted the progression event by logistic regression model. Fivefold cross-validation, performed by the caret (v6.0-94) package (Kuhn, 2008), was employed to optimize model parameters. The result was evaluated by ROC curve under pROC (v1.18.4) package (Robin et al., 2011). Principal Component Analysis (PCA) and corresponding plots were generated using the prcomp function and ggplot2 (v3.4.3) in R (Wickham, 2016). Furthermore, a risk-score based on the signature was defined by the regression coefficient of this prediction model and relevant the methylated level (range, 0-1) of each DMR.

Due to the absence of relevant prognostic time for each patient, the OR was used instead of the hazard ratio (HR) to evaluate the clinical value of each DMR in the signature, as well as the risk score, GS, and surgical T stage in progression estimation. Both univariate and multivariate analyses were conducted.

## Acknowledgements
All authors sincerely appreciate the patients who contributed to this study.

## Competing interests
C.A.M. is an Academic Editor of Biology Open. C.A.M. was not involved in the editorial assessment of this submission. All authors declare no competing interests.

## Author contributions
Conceptualization: C.A.M., M.L., J.W.; Data curation: A.L., E.R.; Formal analysis: M.L.; Funding acquisition: C.A.M.; Investigation: M.L., J.W.; Methodology: M.L., J.W., A.L.; Project administration: C.A.M.; Supervision: C.A.M.; Visualization: M.L.; Writing – original draft: M.L.; Writing – review & editing: C.A.M., M.L., J.W., A.L., E.R.

## Funding
We gratefully acknowledge support from the PCRP Prostate Cancer Biorepository Network (W81XWH-18-2-0019). Deposited in PMC for immediate release.

## Data and resource availability
The main relevant data and details of resources can be found within the article and supplementary information. Raw sequence data have been deposited in Gene Expression Omnibus (GSE308050). Processed data files and codes to generate all figures and conclusions are available at github repository: https://github.com/ChrisMaherLab/PRAD_EMSeq.

## Peer review history
The peer review history is available online at https://journals.biologists.com/bio/lookup/doi/10.1242/bio.062281.reviewer-comments.pdf

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
