## [Peer Review File · Biology Open]

Methylation-based signature to distinguish indolent and aggressive prostate cancer

Muheng Liao, Jace Webster, Amy Ly, Emily Rozycki and Christopher Maher
DOI: 10.1242/bio.062281

Editor: Lewis Halsey

Review timeline

Original submission:	23 September 2025
Editorial decision:	29 September 2025
First revision received:	21 October 2025
Editorial decision:	24 October 2025
Second revision received:	1 November 2025
Editorial decision:	6 November 2025
Third revision received:	9 November 2025
Accepted:	17 November 2025

Original submission

First decision letter

MS ID#: bio.062281

MS Title: Methylation-based signature to distinguish indolent and aggressive prostate cancer

Authors: Christopher Maher, Muheng Liao, Jace Webster, Amy Ly and Emily Rozycki

I have now reached a decision on the above manuscript - and it only took 7 working days, thanks to our new fast & fair peer review initiative, which you can learn more about here: <https://journals.biologists.com/bio/pages/fast-fair>

The reviewer reports are shown at the bottom of this email or can be accessed, together with a copy of this decision letter, by going to:

As you will see, the reviewers gave favourable reports, but raised some critical points that will require amendments to your manuscript. I hope that you will be able to carry these out, because we would like to be able to accept your paper.

I should be grateful if you would also provide a point-by-point response detailing how you have dealt with the points raised by the reviewers in the 'Response to Reviewers' box. Please attend to all of the reviewers' comments. If you do not agree with any of their criticisms or suggestions please explain clearly why this is so. The reviewer comments appear pretty straightforward - please pay special attention to comments regarding transparency, such as explicitly

defining/justifying low, intermediate and high classification, and making sure that statistical tests used in each figure are clearly described.

Reviewer 2 noted "the data are challenging to evaluate due to small text even when the PDF is magnified 200%. Possibly Figure 1A and 1B could be provided as supplementary data and then 1C-E could be increased in size with larger text. In Figure 1D, the yellow versus blue bars are not labeled, presumably yellow = high risk and blue is low-risk. Similarly, Figure 2 is challenging to assess at 100% magnification." For the revised manuscript, please make sure to upload a separate high resolution file for the figures, so we can assess the data at high magnification and read the text easily.

At this stage, we also ask you to ensure your manuscript complies with our formatting guidelines - please see our manuscript preparation guidelines for details. Provided you are able to fully address the referees' comments, we are positive about publication of your paper (we accept over 95% of revision submissions) and therefore hope you won't mind any extra work involved in reformatting your manuscript at this point.

Reviewer 1

Comments for the author

Liao and colleagues used whole-genome enzymatic methylation sequencing of a 120 prostate cancer patient cohort to identify a panel of 14 differentially-methylated regions that effectively predicts the likelihood of aggressive prostate cancer progression.

The Authors began by providing useful quality-control analyses of their EM-seq data, demonstrating consistent patterns of CpG detection across annotated genomic regions and CpG islands among their cohort. Further benchmarking was performed through analysis of DMRs between low- and high-risk groups, revealing improved DMR detection using EM-seq and stratification of patients in each risk group as well as DMRs in the known PCa biomarker genes, APC and GSTP1. GO term enrichment and GSEA was then assessed for the low- and high-risk DMRs, revealing GO and GSEA terms associated with cell motility and signaling.

After benchmarking, the Authors used a computational workflow including Elastic Net Regression, XGBoost, Random Forest, and logistic regression with LASSO penalty to identify a 14-DMR panel for stratifying PCa patients by risk. The Authors then used this panel to compute risk scores, which proved robust at predicting tumor progression (but not PSA recurrence, lymph node metastasis, and mortality) in intermediate- and high-risk samples using a variety of analyses. Further, the 14-DMR panel was used to stratify patients, as determined using hierarchical clustering and PCA, revealing potential disease subsets.

Overall, I believe that this was a well-executed study that is scientifically sound and of general interest to the cancer biology community. The authors demonstrated improved predictive capabilities over standard pathological approaches, and improved ability to use EM-seq for clinical methylome profiling versus alternative techniques. I recommend that the work be published in Biology Open, although I have minor comments that may improve readability and interpretation.

Comments:

1. Authors should consider repeating their GO term enrichment and GSEA analyses after sample label permutation before definitively linking the observed results to high-risk prostate cancer.

2. How did the authors define gene associations for the 14-DMR panel? Within the gene? Within 1kb of the gene?
3. "Similarly, within CpG island annotations, 7.7% were located on islands, 7.2% on shores, and 4.2% on shelves (Fig 1B)." Aid the reader by defining island, shore, and shelves
4. "Similarly, Chr13 had the highest coverage proportion, while ChrY had the lowest (ChrY might got relatively minimal association with tumor risk) ... These groups were well separated and clustered." Awkward phrasing.
5. "...reinforcing the role of membrane activity in aggressive prostate cancer." What do the authors mean by 'membrane activity'? Maybe replace with differential motility and signaling? Is there anything known about EGF signaling in aggressive PCa?
6. It makes sense that the low- and high-risk groups would form distinct clades in the hierarchical clustering results shown in Fig. 1E since the authors are using DRMs that were found by comparing low- and high-risk groups. However, is it surprising that all of the intermediate-risk samples clustered separately, as well?
7. Provide more details about the annotations for Fig. 1C - what are the grey boxes? Not explained in the figure legend.
8. Y-axis text for bottom box-and-whisker plot needs reformatting.
9. Provide more details for Fig. 1E in the legend - Is 'methylation value" a z-score? What are the units for the 'methylation position' box and whisker plot?
10. No p-value provided in Fig. S4B.
11. What statistical test was used for Fig. S8?
12. Authors should provide access to processed data files and code used to generate all figures and conclusions.

Reviewer 2

Comments for the author

The manuscript from Liao et al. entitled, "Methylation-based signature to distinguish indolent and aggressive prostate cancer," aims to define a methylation signature based on the analysis a retrospective cohort of 120 prostate cancer patients. The authors use a enzymatic methylation sequencing (EM-seq) pipeline that allows for assessment of global methylation, which is advantageous over array-based platforms.

Overall, the experimental quality is considered quite high with 120 samples processed for EM-seq, but several details are missing to fully evaluate this criteria. While the authors indicate what methods were used to assess sample purity, they do not indicate a cutoff value for which samples would be excluded, or if any samples did not meet quality control measures. This is true for analysis of the genomic DNA and library preparation.

Note, as presented in the figures, the data are challenging to evaluate due to small text even when the PDF is magnified 200%. Possibly Figure 1A and 1B could be provided as supplementary data and then 1C-E could be increased in size with larger text. In Figure 1D, the yellow versus blue bars are not labeled, presumably yellow = high risk and blue is low-risk. Similarly, Figure 2 is challenging to assess at 100% magnification.

The terms 'middle' and 'risky' is used within Figure 2 and supplementary Figures, but not within the text.

There are several issues with reproducibility of the presented data. The genes associated with the 200 DMR presented in Figure 1E are not listed anywhere, therefore it would be impossible to reproduce data within Figure 1E and 1D. Further is it presumed that not all DMR regions were associated with a gene, as several were not within the 14 gene panel.

The duration of follow-up (average or range) is not indicated. PSA at time of diagnosis is often included in risk stratification, but it is unclear if this data is available for the cohort.

One potential flaw in the experimental design is that it appears the authors do not use NCCN guidelines for establishing low, intermediate and high-risk populations, and the authors do not detail the rationale for the chosen low, intermediate and high classification. This is a significant issue as it will be difficult to apply the results of these studies to other patient populations. Minimally, this needs to be acknowledge.

How the results presented here relate to prior publications stratifying intermediate-high vs. intermediate-low risk groups is not discussed (for example, PMID: 30153435).

Reviewer's Responses to Questions

Experimental quality

Does each figure have the proper controls?

If 'No', please indicate reasons in Comments for Author box below.

Reviewer #1:

- Yes

Reviewer #2:

- Yes

Were the data analyzed using appropriate statistical tests?

If 'No', please indicate reasons in Comments for Author box below.

Reviewer #1:

- Yes

Reviewer #2:

- Yes

Reproducibility

Were experiments performed using adequate number of biological replicates?

If 'No', please indicate reasons in Comments for Author box below.

Reviewer #1:

- Yes

Reviewer #2:

- Yes

Does the methods section provide sufficient detail to permit reproducibility?

If 'No', please indicate reasons in Comments for Author box below.

Reviewer #1:

- Yes

Reviewer #2:

- No

Completeness

Are the manuscript's conclusions supported by the data?

If 'No', please indicate reasons in Comments for Author box below.

Reviewer #1:

- Yes

Reviewer #2:

- Yes

Scholarship

Do the authors cite and discuss the merits of data that would argue for and against their conclusion?

If 'No', please indicate reasons in Comments for Author box below.

Reviewer #1:

- Yes

Reviewer #2:

- No

Does the manuscript title & abstract accurately reflect the contents of the manuscript, without hyperbole?

If 'No', please indicate reasons in Comments for Author box below.

Reviewer #1:

- Yes

Reviewer #2:

- Yes

First revision

Author response to reviewers' comments

Comments:

1. Authors should consider repeating their GO term enrichment and GSEA analyses after sample label permutation before definitively linking the observed results to high-risk prostate cancer.
 - We appreciate this suggestion. Performing the sample label permutation would add another layer of robustness to the analysis. However this would be computationally intensive since it would require us to repeat the methylation analysis, machine learning pipeline, and GSEA analyses multiple times. The current results follow the recommended and widely used GSEA statistical tests for enrichment analysis.

2. How did the authors define gene associations for the 14-DMR panel? Within the gene? Within 1kb of the gene?

We defined gene associations for the 14-DMR panel when the DMR directly overlapped the gene region in the reference annotation (Ensembl release 111). Ensembl groups transcripts into genes with a gene's coordinates running from the 5'-most start to the 3'-most end among its transcripts on a strand. We have clarified this in the **METHODS** in the 'DNA methylation analysis' section as follows:

“DMRs were linked to genes based on direct overlap with the annotated gene region in the reference (Ensembl release 111) (Harrison et al., 2024). DMRs overlapping multiple genes were labeled with each overlapping gene.”

3. “Similarly, within CpG island annotations, 7.7% were located on islands, 7.2% on shores, and 4.2% on shelves (Fig 1B).” Aid the reader by defining island, shore, and shelves
 - We appreciate this suggestion and have clarified this in the Methods within the 'DNA methylation analysis' section as follows:

“The coordinates of CpG islands followed the classic definition from Gardiner-Garden & Frommer's work (Gardiner-Garden and Frommer, 1987). CpG shores were defined as 2kb up- and downstream of islands, excluding the islands. Similarly, CpG shelves were defined as 2kb up- and downstream of shores, excluding shores and islands.”

4. “Similarly, Chr13 had the highest coverage proportion, while ChrY had the lowest (ChrY might got relatively minimal association with tumor risk) ... These groups were well separated and clustered.” Awkward phrasing.
 - We appreciate the suggestion and have updated the text. First, we removed the clause “ChrY might got relatively minimal association with tumor risk”. Second, we revised the Results within the 'Whole-Genome methylation analysis identifies differential methylated regions between low-risk and high-risk prostate cancer patients' section as follows:

“Analyzing DMR total length and coverage proportion on each chromosome (Fig 1D), Chr1 had the longest total DMR length whereas ChrY had the shortest total DMR length. Chr13 had the highest coverage proportion, while ChrY had the lowest. The methylation profiling of the top 200 DMRs with the greatest absolute differences was presented in Fig 1E, illustrating distinct methylation patterns between low- and high-risk groups. The two groups were well separated and clustered.”

5. “...reinforcing the role of membrane activity in aggressive prostate cancer.” What do the authors mean by ‘membrane activity’? Maybe replace with differential motility and signaling? Is there anything known about EGF signaling in aggressive PCa?
- We have updated the results in the ‘Whole-Genome methylation analysis identifies differential methylated regions between low-risk and high-risk prostate cancer patients’ section to clarify membrane activity as follows:
“These results reflected the differences in cell motility and organ morphogenesis between aggressive and indolent prostate tumors (Chen et al., 2023; Clark et al., 2023; Fu et al., 2024). Gene Set Enrichment Analysis (GSEA) further identified two significant gene sets: (a) genes transiently induced by EGF promoting cell-cycle progression; and (b) signal transmission across the membrane through G-protein activation enhancing the exchange of GDP for GTP on the alpha subunit of a heterotrimeric G-protein complex (Fig S4)”.
 - There have been multiple studies focusing on EGF signaling in aggressive prostate cancer.
 Here we highlight a few examples:
 1. Advanced PCa could be promoted by upregulating EGFR ligands. ADAM17 increases their availability, driving persistent EGFR activation—supporting proliferation, migration, and invasion (PMID: 3599805).
 2. EGFR forms heterodimers with ERBB2/ERBB3 that more potently engage PI3K- AKT and MAPK cascades; Dual targeting of EGFR+HER2/HER3 shows stronger preclinical activity than EGFR alone (PMID: 33918389).
 3. High EGFR expression in primary tumors and circulating tumor cells associates with higher grade, shorter metastasis-free survival, and worse prognosis (PMID: 32901137).
6. It makes sense that the low- and high-risk groups would form distinct clades in the hierarchical clustering results shown in Fig. 1E since the authors are using DRMs that were found by comparing low- and high-risk groups. However, is it surprising that all of the intermediate-risk samples clustered separately, as well?
- We appreciate this observation and question. We hypothesized that by comparing the low- and high-risk samples we would identify the discriminatory features for stratifying these different patient populations. Therefore, we were encouraged to see that applying these features to the intermediate-risk patient samples resulted in distinct clusters.
7. Provide more details about the annotations for Fig. 1C - what are the grey boxes? Not explained in the figure legend.
- We have revised Figure 1C to provide more details about the annotations. The grey boxes are centromeres.
8. Y-axis text for bottom box-and-whisker plot needs reformatting.
- We have now reformatting the box-and-whisker plot.

9. Provide more details for Fig. 1E in the legend - Is 'methylation value' a z-score? What are the units for the 'methylation position' box and whisker plot?
- We have provided additional details in Figure 1E.
 - The 'methylation value' is the original proportion of $\frac{\text{reads of unmodified cytosine}}{\text{all reads}} \times 100\%$ in certain site or region.
 - The units for the 'methylation position' box and whisker plot were percentage corresponding to methylation level.
10. No p-value provided in Fig. S4B.
- We have updated Fig. S4B to provide the statistics.
11. What statistical test was used for Fig. S8?
- The statistical test is Wilcoxon signed-rank test. We now provide a detailed explanation of the significance in the figure legend as follows:
*"Fig S8. Methylation comparison of the 14 DMRs within different clinical outcomes (Wilcoxon signed-rank test was applied to get the significance: ns: p-value > 0.05, *: 0.05 > p-value < 0.01, **: 0.01 > p-value > 0.001, ***: 0.001 > p-value)."*
12. Authors should provide access to processed data files and code used to generate all figures and conclusions.
- All data has been published on Gene expression omnibus (GEO) with the accession number: GSE308050.
 - Processed data files and code to generate all figures and conclusions are available at the Maher lab github: https://github.com/ChrisMaherLab/PRAD_EMSeq.

Second decision letter

MS ID#: bio.062281R1

MS Title: Methylation-based signature to distinguish indolent and aggressive prostate cancer

Authors: Christopher Maher, Muheng Liao, Jace Webster, Amy Ly and Emily Rozycki

We received your revised manuscript. I see that you responded to the comments of reviewer 1, but I do not see any response to reviewer 2. I cannot render a decision until you respond to all reviewer comments. I've appended the comments from reviewer 2 again below, and hope you will be able to provide a response and revise the manuscript accordingly.

Please upload both a 'clean' version of your Word file, along with a highlighted version clearly showing where you have made changes in the revised manuscript - this highlighted version should include where you made changes in response to reviewer 1 and reviewer 2. Please avoid using 'Track changes' in Word files as these are lost in PDF conversion.

I should be grateful if you would also provide a point-by-point response detailing how you have dealt with the points raised by the reviewers (both reviewer 1 and reviewer 2 in the same document) in the 'Response to Reviewers' box. Please attend to all of the reviewers' comments. If you do not agree with any of their criticisms or suggestions please explain clearly why this is so.

Second revision

Author response to reviewers' comments

Reviewer 1's comments:

1. Authors should consider repeating their GO term enrichment and GSEA analyses after sample label permutation before definitively linking the observed results to high-risk prostate cancer.
 - We appreciate this suggestion. Performing the sample label permutation would add another layer of robustness to the analysis. However this would be computationally intensive since it would require us to repeat the methylation analysis, machine learning pipeline, and GSEA analyses multiple times. The current results follow the recommended and widely used GSEA statistical tests for enrichment analysis.
2. How did the authors define gene associations for the 14-DMR panel? Within the gene? Within 1kb of the gene?

We defined gene associations for the 14-DMR panel when the DMR directly overlapped the gene region in the reference annotation (Ensembl release 111). Ensembl groups transcripts into genes with a gene's coordinates running from the 5'-most start to the 3'-most end among its transcripts on a strand. We have clarified this in the METHODS within the 'DNA methylation analysis' section as follows:

"DMRs were linked to genes based on direct overlap with the annotated gene region in the reference (Ensembl release 111)(Harrison et al., 2024). DMRs overlapping multiple genes were labeled with all overlapping genes, while not all DMR would be linked to a gene since they might also lay on the non-coding region."
3. "Similarly, within CpG island annotations, 7.7% were located on islands, 7.2% on shores, and 4.2% on shelves (Fig 1B)." Aid the reader by defining island, shore, and shelves
 - We appreciate this suggestion and have clarified this in the METHODS within the 'DNA methylation analysis' section as follows:

"The coordinates of CpG islands followed the classic definition from Gardiner-Garden & Frommer's work (Gardiner-Garden and Frommer, 1987). CpG shores were defined as 2kb up- and downstream of islands, excluding the islands. Similarly, CpG shelves were defined as 2kb up- and downstream of shores, excluding shores and islands."
4. "Similarly, Chr13 had the highest coverage proportion, while ChrY had the lowest (ChrY might got relatively minimal association with tumor risk) ... These groups were well separated and clustered." Awkward phrasing.
 - We appreciate the suggestion and have updated the text. First, we removed the clause "ChrY might got relatively minimal association with tumor risk". Second, we revised the RESULTS within the 'Whole-Genome methylation analysis identifies differential methylated regions between low-risk and high-risk prostate cancer patients' section as follows:

"Analyzing DMR total length and coverage proportion on each chromosome

(Fig 1D), Chr1 had the longest total DMR length whereas ChrY had the shortest total DMR length. Chr13 had the highest coverage proportion, while ChrY had the lowest.

The methylation profiling of the top 200 DMRs with the greatest absolute differences was presented in Fig 1E, illustrating distinct methylation patterns between low- and high-risk groups. The two groups were well separated and clustered.”

5. “...reinforcing the role of membrane activity in aggressive prostate cancer.” What do the authors mean by ‘membrane activity’? Maybe replace with differential motility and signaling? Is there anything known about EGF signaling in aggressive PCa?

- We have updated the RESULTS within the ‘Whole-Genome methylation analysis identifies differential methylated regions between low-risk and high-risk prostate cancer patients’ section to clarify membrane activity as follows:

“These results reflected the differences in cell motility and organ morphogenesis between aggressive and indolent prostate tumors (Chen et al., 2023; Clark et al., 2023; Fu et al., 2024). Gene Set Enrichment Analysis (GSEA) further identified two significant gene sets: (a) genes transiently induced by EGF promoting cell-cycle progression; and (b) signal transmission across the membrane through G-protein activation enhancing the exchange of GDP for GTP on the alpha subunit of a heterotrimeric G-protein complex (Fig S4)”.

- There have been multiple studies focusing on EGF signaling in aggressive prostate cancer.

Here we highlight a few examples:

1. Advanced PCa could be promoted by upregulating EGFR ligands. ADAM17 increases their availability, driving persistent EGFR activation—supporting proliferation, migration, and invasion (PMID: 3599805).
2. EGFR forms heterodimers with ERBB2/ERBB3 that more potently engage PI3K- AKT and MAPK cascades; Dual targeting of EGFR+HER2/HER3 shows stronger preclinical activity than EGFR alone (PMID: 33918389).
3. High EGFR expression in primary tumors and circulating tumor cells associates with higher grade, shorter metastasis-free survival, and worse prognosis (PMID: 32901137).

6. It makes sense that the low- and high-risk groups would form distinct clades in the hierarchical clustering results shown in Fig. 1E since the authors are using DRMs that were found by comparing low- and high- risk groups. However, is it surprising that all of the intermediate-risk samples clustered separately, as well?

- We appreciate this observation and question. We hypothesized that by comparing the low- and high-risk samples we would identify the discriminatory features for stratifying these different patient populations. Therefore, we were encouraged to see that applying these features to the intermediate-risk patient samples resulted in distinct clusters. And we have clarified this information in the section “Whole-Genome methylation analysis identifies differential methylated regions between low-risk and high-risk prostate cancer patients” in RESULTS:

“The methylation profiling of the top 200 DMRs with the greatest absolute differences was presented in Fig 1E, illustrating distinct methylation

patterns between low- and high-risk groups. Two groups were well separated and clustered. While the intermediate-risk patients might have the feature of either group, it implied the possibility and necessity of introducing subgroups.”

7. Provide more details about the annotations for Fig. 1C - what are the grey boxes? Not explained in the figure legend.
 - We have revised Figure 1C to provide more details about the annotations. The grey boxes are centromeres.
8. Y-axis text for bottom box-and-whisker plot needs reformatting.
 - We have now reformatted the box-and-whisker plot.
9. Provide more details for Fig. 1E in the legend - Is ‘methylation value’ a z-score? What are the units for the ‘methylation position’ box and whisker plot?
 - We have provided additional details in Figure 1E.
 - The ‘methylation value’ is the original proportion of $\frac{\text{reads of unmodified cytosine}}{\text{all reads}} \times 100\%$ in certain site or region.
 - The units for the ‘methylation position’ box and whisker plot were percentage corresponding to methylation level.
10. No p-value provided in Fig. S4B.
 - We have updated Fig. S4B to provide the statistics.
11. What statistical test was used for Fig. S8?
 - The statistical test is Wilcoxon signed-rank test. We now provide a detailed explanation of the significance in the figure legend as follows:
*“Fig S8. Methylation comparison of the 14 DMRs within different clinical outcomes (Wilcoxon signed-rank test was applied to get the significance: ns: p-value > 0.05, *: 0.05 > p-value < 0.01, **: 0.01 > p-value > 0.001, ***: 0.001 > p-value).”*
12. Authors should provide access to processed data files and code used to generate all figures and conclusions.
 - All data has been published on Gene expression omnibus (GEO) with the accession number: GSE308050.
 - Processed data files and code to generate all figures and conclusions are available at the Maher lab github:
https://github.com/ChrisMaherLab/PRAD_EMSeq.

Reviewer 2’s comments:

The manuscript from Liao et al. entitled, "Methylation-based signature to distinguish indolent and aggressive prostate cancer," aims to define a methylation signature based on the analysis a retrospective cohort of 120 prostate cancer patients. The authors use a enzymatic methylation sequencing (EM-seq) pipeline that allows for assessment of global methylation, which is advantageous over array-based platforms.

Overall, the experimental quality is considered quite high with 120 samples processed for EM-seq,

but several details are missing to fully evaluate this criteria. While the authors indicate what methods were used to assess sample purity, they do not indicate a cutoff value for which samples would be excluded, or if any samples did not meet quality control measures. This is true for analysis of the genomic DNA and library preparation.

- DNA concentration ≥ 20 ng/ μ L, A260/A280 ratio between 1.8-2.0, and A260/A230 ratio ≥ 1.8 are the cutoff values we used in this study. And all samples we discussed here met these criteria. We added this information into section ‘Enzymatic methylation sequencing experiment’ in METHODS as follows:

“Genomic DNA was extracted using Zymo Research Quick-DNA/RNA MiniPrep kit (Zymo Research, Irvine, CA, USA) and quantified by Qubit 2.0 fluorometer (Thermo Fisher Scientific, Waltham, MA, USA). Sample purity was determined by using the Nanodrop ND-1000 Spectrophotometer (Thermo Fisher Scientific, Waltham, MA, USA). And all samples were required to meet the following quality thresholds prior to library construction: DNA concentration ≥ 20 ng/ μ L, A260/A280 ratio between 1.8-2.0, and A260/A230 ratio ≥ 1.8 .”

Note, as presented in the figures, the data are challenging to evaluate due to small text even when the PDF is magnified 200%. Possibly Figure 1A and 1B could be provided as supplementary data and then 1C-E could be increased in size with larger text. In Figure 1D, the yellow versus blue bars are not labeled, presumably yellow = high risk and blue is low-risk. Similarly, Figure 2 is challenging to assess at 100% magnification.

- We have adjusted the font size of Figure 1&2. For Figure 1D, the yellow bars referred to left yellow Y axis which was proportion of DMRs on each chromosome, while the blue bars referred to right blue Y axis corresponding to total length of DMRs. We have added a legend and adjusted the color of axis title and figure description to make it clearer.

The terms ‘middle’ and ‘risky’ is used within Figure 2 and supplementary Figures, but not within the text.

- We have revised these terms from “middle-risk” to “intermediate-risk”. This has been updated in the text.

There are several issues with reproducibility of the presented data. The genes associated with the 200 DMR presented in Figure 1E are not listed anywhere, therefore it would be impossible to reproduce data within Figure 1E and 1D. Further is it presumed that not all DMR regions were associated with a gene, as several were not within the 14 gene panel.

- To allow for reproducibility, the processed data has been published on Maher lab github: https://github.com/ChrisMaherLab/PRAD_EMSeq. This includes the genes associated with the 200 DMRs.
- Yes, not all DMRs would be associated with a gene. It depended on whether they could directly overlap the gene region in the reference annotation (Ensembl release 111). Ensembl groups transcripts into genes with a gene’s coordinates running from the 5’-most start to the 3’-most end among its transcripts on a strand. Any DMR located outside of those regions would be referred to as no gene. We have clarified this in the section of “DNA methylation analysis” in METHODS:

“DMRs were linked to genes based on direct overlap with the annotated gene region in the reference (Ensembl release 111)(Harrison et al., 2024). DMRs overlapping multiple genes were labeled with all overlapping genes, while not all DMR would be linked to a gene since they might also lay on the non-coding region.”

The duration of follow-up (average or range) is not indicated. PSA at time of diagnosis is often included in risk stratification, but it is unclear if this data is available for the cohort.

- We appreciate this suggestion, but unfortunately this data is not available for the cohort.

One potential flaw in the experimental design is that it appears the authors do not use NCCN guidelines for establishing low, intermediate and high-risk populations, and the authors do not detail the rationale for the chosen low, intermediate and high classification. This is a significant issue as it will be difficult to apply the results of these studies to other patient populations. Minimally, this needs to be acknowledge.

- We appreciate this suggestion. NCCN Guidelines is an important reference for the risk stratification.

We made some adjustments for a good contrast between low- and high-risk. We have updated this information in the METHODS within “Study population and sample collection” section:

“Clinical outcomes were followed until loss connection, with tumor progression defined by the occurrence of any of the following: death, tumor recurrence, PSA recurrence, or lymph node metastasis. To stratify the degree of malignancy of the tumor, risk stratification criteria from NCCN Guidelines were utilized and subsequently optimized to achieve a greater contrast between low- and high-risk patients while maintaining balanced group sizes (Schaeffer et al., 2024). Patients with GS = 7 and surgical T stage $\leq 2c$ were classified as low-risk, those with GS ≥ 9 and surgical T stage $\geq 3b$ as high-risk, and the remaining patients as intermediate-risk. All samples were formalin-fixed and paraffin-embedded (FFPE) in 4°C environment.”

How the results presented here relate to prior publications stratifying intermediate-high vs. intermediate-low risk groups is not discussed (for example, PMID: 30153435).

- We appreciate this suggestion and have updated our DISCUSSION section to address this as follows:

“During clinical prognosis, several traditional markers of prostate cancer have been widely applied. Gleason score serves as the cornerstone for histological grading, while tumor T stage provides essential anatomical staging information (Egevad et al., 2002; Mo et al., 2023; Egevad et al., 2002). These parameters have been integral to clinical decision-making in NCCN Guidelines for decades. However, their limitations in accurately predicting disease progression, particularly in intermediate-risk patients, have been increasingly recognized. Several groups have proposed subdividing intermediate-risk prostate cancer into favorable and unfavorable categories based on factors such as primary Gleason score, number of intermediate-risk features, and percentage of positive biopsy cores (Courtney et al., 2022; Zumsteg et al., 2013). These refined classifications improved prognostic discrimination for biochemical

recurrence, metastasis, and mortality compared with the traditional three-tier model. Additionally, proper treatment also requires further stratification in the intermediate-risk group. For instance, prior works suggests that it is unlikely that treatment intensification would meaningfully improve oncologic outcomes in favorable intermediate risk group (Berlin et al., 2019). However, over- or under-treatment could be a potential problem. Our findings were consistent with and extend this concept at the molecular level. Within our cohort, the majority of patients (62.5%) fell into the intermediate-risk group, which historically exhibits the greatest uncertainty in treatment selection. The strong discriminative ability of our 14-DMR methylation signature effectively separated progressive from indolent cases within this heterogeneous population, paralleling the clinical distinction between unfavorable and favorable intermediate-risk disease. In this situation, our 14-DMR signature demonstrated markedly superior predictive accuracy (AUC = 0.92) and sharply contrasted with Gleason score (AUC = 0.65) and tumor T stage (AUC = 0.55). This substantial improvement represents a clinically meaningful advancement. The intermediate-risk category made up 62.5% of this cohort and faced the greatest uncertainty regarding treatment selection and prognosis. The enhanced performance of our methylation signature highlights its potential clinical utility.”

Third decision letter

MS ID#: bio.062281R2

MS Title: Methylation-based signature to distinguish indolent and aggressive prostate cancer

Authors: Christopher Maher, Muheng Liao, Jace Webster, Amy Ly and Emily Rozycki

I have now reached a decision on the above manuscript.

I appreciate you taking the time to respond to the reviewer comments. The comments have mostly been addressed satisfactorily. Below are some points to improve the clarity and transparency of the manuscript. If you can address these (they require modifications to the text, but no new experiments) then we should be able to publish your manuscript.

Reviewer 1, comment 6, on the observation that intermediate-risk samples clustered separately from low- and high-risk samples. Please make this observation clear in the manuscript text (it was addressed in the response to reviewer comments document but not that I could find in the revised manuscript).

Please also make sure that intermediate-risk and not middle-risk is used throughout (eg Fig S6, S7). Fig 1C - In the figure legend, could you indicate that the grey boxes are centromeres? This may not be entirely clear from the line indicator in the figure panel itself.

Fig 1D - please have the Y axis legends match the figure legend text, eg left Y axis could be DMR proportion (%), the right Y DMR length (bp) or similar

Fig 1E - please provide definition of methylation value (%) in the figure legend (I see that you answered the reviewer's question in the response to reviewers document, but I don't see this information in the revised manuscript).

Clarify that not all DMRs are associated with a gene - I see this clarification in the response to reviewer comment but not anywhere in the manuscript text. Perhaps it would help to add a supplementary table listing the 200 DMRs and their gene association (or noting that they are not associated with a gene, based on Ensembl grouping), rather than relying only on github deposition? When establishing low vs high risk populations, you revised the manuscript to use “criteria derived from NCCN Guidelines and were subsequently optimized to achieve a greater contrast between low- and high-risk patients while maintaining balanced group sizes...” What does it mean ‘subsequently optimized’? How were these guidelines optimized?

Would it be possible to add a Limitations section to the Discussion, and mention/discuss the following point, and how it might influence the results and/or future studies: PSA at time of diagnosis could not be included in risk stratification because PSA data not available for this cohort. This section can also include any additional limitations you would like to discuss - this is to improve the rigor of the manuscript, we are not looking for reasons to reject your manuscript :)

At this stage, we also ask you to ensure your manuscript complies with our formatting guidelines - please see our manuscript preparation guidelines for details. Provided you are able to fully address the referees’ comments, we are positive about publication of your paper (we accept over 95% of revision submissions) and therefore hope you won’t mind any extra work involved in reformatting your manuscript at this point.

Third revision

Author response to reviewers’ comments

Reviewer 1’s comments:

1. Authors should consider repeating their GO term enrichment and GSEA analyses after sample label permutation before definitively linking the observed results to high-risk prostate cancer.
 - We appreciate this suggestion. Performing the sample label permutation would add another layer of robustness to the analysis. However this would be computationally intensive since it would require us to repeat the methylation analysis, machine learning pipeline, and GSEA analyses multiple times. The current results follow the recommended and widely used GSEA statistical tests for enrichment analysis.

2. How did the authors define gene associations for the 14-DMR panel? Within the gene? Within 1kb of the gene?

We defined gene associations for the 14-DMR panel when the DMR directly overlapped the gene region in the reference annotation (Ensembl release 111). Ensembl groups transcripts into genes with a gene’s coordinates running from the 5’-most start to the 3’-most end among its transcripts on a strand. We have clarified this in the METHODS within the ‘DNA methylation analysis’ section as follows:

“DMRs were linked to genes based on direct overlap with the annotated gene region in the reference (Ensembl release 111)(Harrison et al., 2024). DMRs overlapping multiple genes were labeled with all overlapping genes, while not all DMR would be linked to a gene since they might also lay on the non-coding region.”

3. “Similarly, within CpG island annotations, 7.7% were located on islands, 7.2% on shores,

and 4.2% on shelves (Fig 1B).” Aid the reader by defining island, shore, and shelves

- We appreciate this suggestion and have clarified this in the METHODS within the ‘DNA methylation analysis’ section as follows:

“The coordinates of CpG islands followed the classic definition from Gardiner-Garden & Frommer’s work (Gardiner-Garden and Frommer, 1987). CpG shores were defined as 2kb up- and downstream of islands, excluding the islands. Similarly, CpG shelves were defined as 2kb up- and downstream of shores, excluding shores and islands.”

4. “Similarly, Chr13 had the highest coverage proportion, while ChrY had the lowest (ChrY might got relatively minimal association with tumor risk) ... These groups were well separated and clustered.” Awkward phrasing.

- We appreciate the suggestion and have updated the text. First, we removed the clause “ChrY might got relatively minimal association with tumor risk”. Second, we revised the RESULTS within the ‘Whole-Genome methylation analysis identifies differential methylated regions between low-risk and high-risk prostate cancer patients’ section as follows:

“Analyzing DMR total length and coverage proportion on each chromosome (Fig 1D), Chr1 had the longest total DMR length whereas ChrY had the shortest total DMR length. Chr13 had the highest coverage proportion, while ChrY had the lowest.

The methylation profiling of the top 200 DMRs with the greatest absolute differences was presented in Fig 1E, illustrating distinct methylation patterns between low- and high-risk groups. The two groups were well separated and clustered.”

5. “...reinforcing the role of membrane activity in aggressive prostate cancer.” What do the authors mean by ‘membrane activity’? Maybe replace with differential motility and signaling? Is there anything known about EGF signaling in aggressive PCa?

- We have updated the RESULTS within the ‘Whole-Genome methylation analysis identifies differential methylated regions between low-risk and high-risk prostate cancer patients’ section to clarify membrane activity as follows:

“These results reflected the differences in cell motility and organ morphogenesis between aggressive and indolent prostate tumors (Chen et al., 2023; Clark et al., 2023; Fu et al., 2024). Gene Set Enrichment Analysis (GSEA) further identified two significant gene sets: (a) genes transiently induced by EGF promoting cell-cycle progression; and (b) signal transmission across the membrane through G-protein activation enhancing the exchange of GDP for GTP on the alpha subunit of a heterotrimeric G-protein complex (Fig S4)”.

- There have been multiple studies focusing on EGF signaling in aggressive prostate cancer.

Here we highlight a few examples:

1. Advanced PCa could be promoted by upregulating EGFR ligands. ADAM17 increases their availability, driving persistent EGFR activation—supporting proliferation, migration, and invasion (PMID: 3599805).
2. EGFR forms heterodimers with ERBB2/ERBB3 that more potently engage PI3K- AKT and MAPK cascades; Dual targeting of EGFR+HER2/HER3 shows stronger preclinical activity than EGFR alone (PMID: 33918389).

3. High EGFR expression in primary tumors and circulating tumor cells associates with higher grade, shorter metastasis-free survival, and worse prognosis (PMID: 32901137).

6. It makes sense that the low- and high-risk groups would form distinct clades in the hierarchical clustering results shown in Fig. 1E since the authors are using DRMs that were found by comparing low- and high- risk groups. However, is it surprising that all of the intermediate-risk samples clustered separately, as well?

- We appreciate this observation and question. We hypothesized that by comparing the low- and high-risk samples we would identify the discriminatory features for stratifying these different patient populations. Therefore, we were encouraged to see that applying these features to the intermediate-risk patient samples resulted in distinct clusters. And we have clarified this information in the section “Whole-Genome methylation analysis identifies differential methylated regions between low-risk and high-risk prostate cancer patients” in RESULTS:

“The methylation profiling of the top 200 DMRs with the greatest absolute differences was presented in Fig 1E, illustrating distinct methylation patterns between low- and high-risk groups. Two groups were well separated and clustered. While the intermediate-risk patients might have the feature of either group, it implied the possibility and necessity of introducing subgroups.”

7. Provide more details about the annotations for Fig. 1C - what are the grey boxes? Not explained in the figure legend.

- We have revised Figure 1C to provide more details about the annotations. The grey boxes are centromeres.

8. Y-axis text for bottom box-and-whisker plot needs reformatting.

- We have now reformatted the box-and-whisker plot.

9. Provide more details for Fig. 1E in the legend - Is ‘methylation value’ a z-score? What are the units for the ‘methylation position’ box and whisker plot?

- We have provided additional details in Figure 1E.
- The ‘methylation value’ is the original proportion of $\frac{\text{reads of unmodified cytosine}}{\text{all reads}} \times 100\%$ in certain site or region.
- The units for the ‘methylation position’ box and whisker plot were percentage corresponding to methylation level.

10. No p-value provided in Fig. S4B.

- We have updated Fig. S4B to provide the statistics.

11. What statistical test was used for Fig. S8?

- The statistical test is Wilcoxon signed-rank test. We now provide a detailed explanation of the significance in the figure legend as follows:

*“Fig S8. Methylation comparison of the 14 DMRs within different clinical outcomes (Wilcoxon signed-rank test was applied to get the significance: ns: p-value > 0.05, *: 0.05 > p-value < 0.01, **: 0.01 > p-value > 0.001, ***: 0.001 > p-value).”*

12. Authors should provide access to processed data files and code used to generate all figures and conclusions.

- All data has been published on Gene expression omnibus (GEO) with the accession number: GSE308050.
- Processed data files and code to generate all figures and conclusions are available at the Maher lab github:
https://github.com/ChrisMaherLab/PRAD_EMSeq.

Reviewer 2's comments:

The manuscript from Liao et al. entitled, "Methylation-based signature to distinguish indolent and aggressive prostate cancer," aims to define a methylation signature based on the analysis a retrospective cohort of 120 prostate cancer patients. The authors use a enzymatic methylation sequencing (EM-seq) pipeline that allows for assessment of global methylation, which is advantageous over array-based platforms.

Overall, the experimental quality is considered quite high with 120 samples processed for EM-seq, but several details are missing to fully evaluate this criteria. While the authors indicate what methods were used to assess sample purity, they do not indicate a cutoff value for which samples would be excluded, or if any samples did not meet quality control measures. This is true for analysis of the genomic DNA and library preparation.

- DNA concentration ≥ 20 ng/ μ L, A260/A280 ratio between 1.8-2.0, and A260/A230 ratio ≥ 1.8 are the cutoff values we used in this study. And all samples we discussed here met these criteria. We added this information into section 'Enzymatic methylation sequencing experiment' in METHODS as follows:

"Genomic DNA was extracted using Zymo Research Quick-DNA/RNA MiniPrep kit (Zymo Research, Irvine, CA, USA) and quantified by Qubit 2.0 fluorometer (Thermo Fisher Scientific, Waltham, MA, USA). Sample purity was determined by using the Nanodrop ND-1000 Spectrophotometer (Thermo Fisher Scientific, Waltham, MA, USA). And all samples were required to meet the following quality thresholds prior to library construction: DNA concentration ≥ 20 ng/ μ L, A260/A280 ratio between 1.8-2.0, and A260/A230 ratio ≥ 1.8 ."

Note, as presented in the figures, the data are challenging to evaluate due to small text even when the PDF is magnified 200%. Possibly Figure 1A and 1B could be provided as supplementary data and then 1C-E could be increased in size with larger text. In Figure 1D, the yellow versus blue bars are not labeled, presumably yellow = high risk and blue is low-risk. Similarly, Figure 2 is challenging to assess at 100% magnification.

- We have adjusted the font size of Figure 1&2. For Figure 1D, the yellow bars referred to left yellow Y axis which was proportion of DMRs on each chromosome, while the blue bars referred to right blue Y axis corresponding to total length of DMRs. We have added a legend and adjusted the color of axis title and figure description to make it clearer.

The terms 'middle' and 'risky' is used within Figure 2 and supplementary Figures, but not within the text.

- We have revised these terms from "middle-risk" to "intermediate-risk". This has been updated in the text.

There are several issues with reproducibility of the presented data. The genes associated with the 200 DMR presented in Figure 1E are not listed anywhere, therefore it would be impossible to reproduce data within Figure 1E and 1D. Further is it presumed that not all DMR regions were associated with a gene, as several were not within the 14 gene panel.

- To allow for reproducibility, the processed data has been published on Maher lab github: https://github.com/ChrisMaherLab/PRAD_EMSeq. This includes the genes associated with the 200 DMRs.
- Yes, not all DMRs would be associated with a gene. It depended on whether they could directly overlap the gene region in the reference annotation (Ensembl release 111). Ensembl groups transcripts into genes with a gene's coordinates running from the 5'-most start to the 3'-most end among its transcripts on a strand. Any DMR located outside of those regions would be referred to as no gene. We have clarified this in the section of "DNA methylation analysis" in METHODS:

"DMRs were linked to genes based on direct overlap with the annotated gene region in the reference (Ensembl release 111)(Harrison et al., 2024). DMRs overlapping multiple genes were labeled with all overlapping genes, while not all DMR would be linked to a gene since they might also lay on the non-coding region."

The duration of follow-up (average or range) is not indicated. PSA at time of diagnosis is often included in risk stratification, but it is unclear if this data is available for the cohort.

- We appreciate this suggestion, but unfortunately this data is not available for the cohort.

One potential flaw in the experimental design is that it appears the authors do not use NCCN guidelines for establishing low, intermediate and high-risk populations, and the authors do not detail the rationale for the chosen low, intermediate and high classification. This is a significant issue as it will be difficult to apply the results of these studies to other patient populations. Minimally, this needs to be acknowledge.

- We appreciate this suggestion. NCCN Guidelines is an important reference for the risk stratification. We made some adjustments for a good contrast between low- and high-risk. We have updated this information in the METHODS within "Study population and sample collection" section:

"Clinical outcomes were followed until loss connection, with tumor progression defined by the occurrence of any of the following: death, tumor recurrence, PSA recurrence, or lymph node metastasis. To stratify the degree of malignancy of the tumor, risk stratification criteria from NCCN Guidelines were utilized and subsequently optimized to achieve a greater contrast between low- and high-risk patients while maintaining balanced group sizes (Schaeffer et al., 2024). Patients with GS = 7 and surgical T stage $\leq 2c$ were classified as low-risk, those with GS ≥ 9 and surgical T stage $\geq 3b$ as high-risk, and the remaining patients as intermediate-risk. All samples were formalin-fixed and paraffin-embedded (FFPE) in 4°C environment."

How the results presented here relate to prior publications stratifying intermediate-high vs. intermediate-low risk groups is not discussed (for example, PMID: 30153435).

- We appreciate this suggestion and have updated our DISCUSSION section to address this as follows:

“During clinical prognosis, several traditional markers of prostate cancer have been widely applied. Gleason score serves as the cornerstone for histological grading, while tumor T stage provides essential anatomical staging information (Egevad et al., 2002; Mo et al., 2023; Egevad et al., 2002). These parameters have been integral to clinical decision-making in NCCN Guidelines for decades. However, their limitations in accurately predicting disease progression, particularly in intermediate-risk patients, have been increasingly recognized. Several groups have proposed subdividing intermediate-risk prostate cancer into favorable and unfavorable categories based on factors such as primary Gleason score, number of intermediate-risk features, and percentage of positive biopsy cores (Courtney et al., 2022; Zumsteg et al., 2013). These refined classifications improved prognostic discrimination for biochemical recurrence, metastasis, and mortality compared with the traditional three-tier model. Additionally, proper treatment also requires further stratification in the intermediate-risk group. For instance, prior works suggests that it is unlikely that treatment intensification would meaningfully improve oncologic outcomes in favorable intermediate risk group (Berlin et al., 2019). However, over- or under-treatment could be a potential problem. Our findings were consistent with and extend this concept at the molecular level. Within our cohort, the majority of patients (62.5%) fell into the intermediate-risk group, which historically exhibits the greatest uncertainty in treatment selection. The strong discriminative ability of our 14-DMR methylation signature effectively separated progressive from indolent cases within this heterogeneous population, paralleling the clinical distinction between unfavorable and favorable intermediate-risk disease.

In this situation, our 14-DMR signature demonstrated markedly superior predictive accuracy (AUC = 0.92) and sharply contrasted with Gleason score (AUC = 0.65) and tumor T stage (AUC = 0.55). This substantial improvement represents a clinically meaningful advancement. The intermediate-risk category made up 62.5% of this cohort and faced the greatest uncertainty regarding treatment selection and prognosis. The enhanced performance of our methylation signature highlights its potential clinical utility.”

Editor’s comments:

Reviewer 1, comment 6, on the observation that intermediate-risk samples clustered separately from low- and high-risk samples. Please make this observation clear in the manuscript text (it was addressed in the response to reviewer comments document but not that I could find in the revised manuscript).

- We have revised the answer and clarified this information in the section “Whole-Genome methylation analysis identifies differential methylated regions between low-risk and high-risk prostate cancer patients” in RESULTS:

“The methylation profiling of the top 200 DMRs with the greatest

absolute differences was presented in Fig 1E, illustrating distinct methylation patterns between low- and high-risk groups. Two groups were well separated and clustered. While the intermediate-risk patients might have the feature of either group, it implied the possibility and necessity of introducing subgroups.”

Please also make sure that intermediate-risk and not middle-risk is used throughout (eg Fig S6, S7).

- We have updated the figure description to make everything consistent in “intermediate-risk”.

Fig 1C - In the figure legend, could you indicate that the grey boxes are centromeres? This may not be entirely clear from the line indicator in the figure panel itself.

- Relevant description was updated in figure description.

Fig 1D - please have the Y axis legends match the figure legend text, eg left Y axis could be DMR proportion (%), the right Y DMR length (bp) or similar

- The titles of axes have been revised to match the figure description.

Fig 1E - please provide definition of methylation value (%) in the figure legend (I see that you answered the reviewer’s question in the response to reviewers document, but I don’t see this information in the revised manuscript).

- A simple definition has been added to figure description.

Clarify that not all DMRs are associated with a gene - I see this clarification in the response to reviewer comment but not anywhere in the manuscript text. Perhaps it would help to add a supplementary table listing the 200 DMRs and their gene association (or noting that they are not associated with a gene, based on Ensembl grouping), rather than relying only on github deposition?

- We have clarified this in the section of “DNA methylation analysis” in METHODS:

“DMRs were linked to genes based on direct overlap with the annotated gene region in the reference (Ensembl release 111)(Harrison et al., 2024). DMRs overlapping multiple genes were labeled with all overlapping genes, while not all DMR would be linked to a gene since they might also lay on the non-coding region.”

When establishing low vs high risk populations, you revised the manuscript to use “criteria derived from NCCN Guidelines and were subsequently optimized to achieve a greater contrast between low- and high-risk patients while maintaining balanced group sizes...”What does it mean ‘subsequently optimized’? How were these guidelines optimized?

- ‘subsequently optimized’ was just the real criteria of risk stratification we used. We have adjusted the expression to make it clearer:

“Clinical outcomes were followed until loss connection, with tumor progression defined by the occurrence of any of the following: death, tumor recurrence, PSA recurrence, or lymph node metastasis. To stratify the

degree of malignancy of the tumor, risk stratification criteria from NCCN Guidelines were utilized and subsequently optimized to achieve a greater contrast between low- and high-risk patients while maintaining balanced group sizes (Schaeffer et al., 2024). Patients with GS = 7 and surgical T stage $\leq 2c$ were classified as low-risk, those with GS ≥ 9 and surgical T stage $\geq 3b$ as high-risk, and the remaining patients as intermediate- risk. All samples were formalin-fixed and paraffin-embedded (FFPE) in 4°C environment.”

Would it be possible to add a Limitations section to the Discussion, and mention/discuss the following point, and how it might influence the results and/or future studies: PSA at time of diagnosis could not be included in risk stratification because PSA data not available for this cohort. This section can also include any additional limitations you would like to discuss - this is to improve the rigor of the manuscript, we are not looking for reasons to reject your manuscript :)

- We appreciated this suggestion and have updated a paragraph at the end of DISCUSSION:

“Nevertheless, this study has several limitations. First, our risk stratification strategy was imperfect. The lack of follow-up time and PSA diagnosis might have led to the misclassification of a small number of patients into the current intermediate-risk category. Additionally, the methylation signature and risk-score held potential for non-invasive detection using cell-free DNA. However, this potential could not be validated within the present study due to the lack of corresponding fluid samples. Therefore, validating their feasibility in liquid biopsies would be the focus of a subsequent independent investigation.”

Fourth decision letter

MS ID#: bio.062281R3

MS Title: Methylation-based signature to distinguish indolent and aggressive prostate cancer

Authors: Christopher Maher, Muheng Liao, Jace Webster, Amy Ly and Emily Rozycki

I am happy to tell you that your manuscript has been accepted for publication in Biology Open, pending our standard publication integrity checks. It was accepted on 17th November 2025.